

# Analysis of the RNA virome of basal hexapods

Sabina Ott Rutar and  Dusan Kordis

Department of Molecular and Biomedical Sciences, Josef Stefan Institute, Ljubljana, Slovenija

## ABSTRACT

The diversity and evolution of RNA viruses has been well studied in arthropods and especially in insects. However, the diversity of RNA viruses in the basal hexapods has not been analysed yet. To better understand their diversity, evolutionary histories and genome organizations, we searched for RNA viruses in transcriptome and genome databases of basal hexapods. We discovered  40 novel RNA viruses, some of which are also present as endogenous viral elements derived from RNA viruses. Here, we demonstrated that basal hexapods host 14 RNA viral clades that have been recently identified in invertebrates. The following RNA viral clades are associated with basal hexapods: Reo, Partiti-Picobirna, Toti-Chryso, Mono-Chu, Bunya-Arena, Orthomyxo, Qinvirus, Picorna-Calici, Hepe-Virga, Narna-Levi, Tombus-Noda, Luteo-Sobemo, Permutotetra and Flavi. We have found representatives of the nine RNA viral clades that are present as endogenous genomic copies in the genomes of Machilis (Monocondylia) and Catajapyx (Diplura). Our study provided a first insight into the diversity of RNA viruses in basal hexapods and demonstrated that the basal hexapods possess quite high diversity of RNA viral clades.

## INTRODUCTION

The analysis of the invertebrate RNA virosphere uncovered a vast diversity of RNA viruses in insects, their evolutionary histories, highly diverse and dynamic genome organizations, as well as the presence of distinct RNA viromes in diverse insect lineages (*Shi et al., 2016*). Metazoans possess the largest diversity of RNA viruses that belong to 23 out of 24 RNA viral clades (*Shi et al., 2016*; *Shi et al., 2018*) and the majority of these data were discovered in arthropods (*Shi et al., 2016*; *Li et al., 2015*). Previously defined virus families, orders, floating genera and novel virus groups were merged together into 24 RNA viral clades. Their names reflect the presence of representative viral families or orders within each RNA viral clade (*Shi et al., 2016*; *Shi et al., 2018*). Some of the RNA viral clades are large and widespread, while the majority of them have a quite limited distribution. As demonstrated in phylogenetic analyses (*Shi et al., 2016*; *Li et al., 2015*), novel data connected plant and animal RNA viral clades, offering a possible interpretation for their dissemination through horizontal transfer by insect vectors (*Shi et al., 2016*; *Li et al., 2015*; *Dolja & Koonin, 2018*; *Blanc & Gutierrez, 2015*). Insects are a rich source of the RNA viral diversity because of their high taxa diversity, omnipresence and ecological interactions with vertebrates and plants.

Corresponding author
Dusan Kordis, dusan.kordis@ijs.si

**How to cite this article** Ott Rutar S, Kordis D. 2020. Analysis of the RNA virome of basal hexapods. *PeerJ* 8:e8336
http://doi.org/10.7717/peerj.8336

Many insects are known vectors for the dissemination of RNA viruses, such as mosquitoes and many plant pests (e.g., thrips, whiteflies, Hemiptera and scale insects) (*Whitfield, Falk & Rotenberg, 2015*; *Rückert & Ebel, 2018*). However, unequal sampling of RNA viruses in diverse taxonomic lineages represents a major problem in the interpretation of their origins and evolution (*Dolja & Koonin, 2018*).

Numerous arthropod groups have not been included in the extensive analyses of RNA viral diversity. One of such neglected taxonomic groups are basal hexapods (the former "apterygote" insects). They are an assemblage of five groups: Protura (coneheads), Collembola (springtails), Diplura (two-pronged bristletails), Monocondylia: Archaeognatha (jumping bristletails) and Zygentoma (bristletails, silverfish and firebrats) and represent the earliest splits of hexapod lineages (*Misof et al., 2014*). Basal hexapods are characterized by their primary lack of wings. Many basal hexapods are of great ecological and economic importance. Especially Collembola play a vital role in soil and leaf litter decomposition (*Rusek, 1998*). Some cosmopolitan species are pests, like *Lepisma saccharina* (Zygentoma) or the lucerna flea *Sminthurus viridis* (Collembola). Springtails have the widest distribution of any hexapod group, occurring throughout the world, including Antarctica. They are found in soil, leaf litter, logs, dung, cave, shorelines, etc. and are probably the most abundant hexapods on Earth, with up to one quarter of billion individuals per square acre (*Rusek, 1998*).

RNA viruses in the basal hexapods have largely been ignored and were not included in the study of invertebrate RNA virosphere (*Shi et al., 2016*). Until now, the only reported basal hexapod RNA virus was an amalgavirus found in a springtail (*Pyle, Keeling & Nibert, 2017*). Therefore, our goal was to gather new information about the distribution and diversity of RNA viruses associated with basal hexapods, the composition of their RNA virome and to compare the basal hexapod RNA virome with the data from the analysed insect orders (*Shi et al., 2016*). Our aim was also to evaluate the potential cases of horizontal transfer of RNA viruses between plants and basal hexapods due to their involvement in the decomposition of plant material. By mining 16 transcriptomes (at the NCBI Transcriptome Shotgun Assembly (TSA) database) and six genomes (at the NCBI Whole Genome Shotgun database) of basal hexapods, we identified genomes of ~40 novel and diverse RNA viruses. Our study provides the first insight into the diversity of RNA viruses and the composition of their RNA virome in basal hexapods. Here, we demonstrated that the RNA virome of basal hexapods is rich and more diverse than that of numerous large insect orders.

## MATERIALS & METHODS

### Discovery of RNA viruses in public transcriptomic databases

Sequence database searches were finished in May 2019. The protein queries were RNA dependent RNA polymerase (RdRp) sequences representing every RNA virus family recognized by ICTV (*Lefkowitz et al., 2018*), as well as the majority of the RNA viruses that are unclassified. The protein queries were also sequences of structural proteins from diverse RNA virus families. The database analysed was the Transcriptome Shotgun Assembly (TSA) at the National Center for Biotechnology Information (http://www.ncbi.nlm.nih.gov). To

detect all available representatives of the particular RNA viral family, database searches were performed iteratively. Comparisons were made using the TBLASTN program (*Gertz et al., 2006*), with the $E$-value cutoff set to $10^{-5}$ and default settings for other parameters. The most divergent representatives of the particular RNA viral family were used as queries. All newly obtained sequences were compared to reference protein sequences of all RNA viruses. Sequences yielding e-values larger than $1e^{-5}$ were retained and compared to entire NCBI NR database to exclude non-viral sequences. Sequences for which the top hit was a virus and sequences with no other BLASTP hits in NCBI NR Db were then treated as putatively viral in origin and subject to further analysis. To detect highly divergent viruses, we performed domain-based BLAST by comparing the newly obtained sequences against the conserved domain database with an expected value threshold of $1 \times 10^{-2}$. Sequences with positive hits to the RdRp domain were retained. DNA sequences were translated with the Translate program (web.expasy.org/translate/). The nucleotide sequences of all basal hexapod RNA viruses are available in the Data S1 file.

## Analysis of endogenous virus elements

Endogenous copies of the RNA viruses were detected using the TBLASTN algorithm against basal hexapod genomes available in the Whole Genome Shotgun Database (WGS) at the NCBI, using viral protein sequences as queries. The queries involved protein sequences translated from both the virus genomes that were identified for the first time here as well as the reference virus genomes. Comparisons were made using the TBLASTN program (*Gertz et al., 2006*), with the $E$-value cutoff set to $10^{-5}$ and default settings for other parameters. For each potential endogenous virus, the query process was reversed to determine their corresponding phylogenetic group. The nucleotide and amino acid sequences of EVEs are available in the Data S1 file.

## Prediction of protein domains

In order to recognize potential protein domains in the protein sequences analysed, we used NCBI CDD database (http://www.ncbi.nlm.nih.gov/Structure/cdd/wrpsb.cgi), by applying a cut-off $E$-value of 0.01. Some proteins were compared against SMART (smart.embl-heidelberg.de), InterPro (http://www.ebi.ac.uk/interpro/) and Pfam (pfam.xfam.org) protein domain databases at default parameters.

## Phylogenetic analysis

To infer the phylogenetic relationships among RNA viruses, we used their RdRp protein sequences. Key representatives of the particular RNA viral family were included in the phylogenetic analysis. The protein sequences of the palm subdomain of RdRps were aligned using MAFFT (*Katoh & Standley, 2013*). Phylogenetic trees were reconstructed using the maximum likelihood (ML) method. For phylogenetic reconstruction, we used IQ-TREE with the in-built automated test to choose the best substitution model for each tree (*Trifinopoulos et al., 2016*). Branch support was computed for all trees using 100 replicates of parametric bootstrap, and 1,000 replicates of the approximate likelihood ratio test and ultrafast bootstrap. The iTOL online tool (http://itol.embl.de/) was used for phylogenetic tree annotation (*Letunic & Bork, 2016*).

## RESULTS

### Discovery of novel and highly divergent RNA viruses in basal hexapods

The collection of the NCBI Transcriptome Shotgun Assembly (TSA) and Whole Genome Sequence (WGS) databases for basal hexapods offers an attractive possibility to obtain the first insight into the diversity of their RNA viromes. We performed the analysis of RNA viruses in 16 transcriptomes and 6 genomes of the springtails (Collembola), silverfish (Zygentoma), diplurans (Diplura) and bristletails (Monocondylia: Archeaognatha) (Table 1). 12 transcriptomes and 2 genomes were positive for RNA viruses, while 4 transcriptomes and 4 genomes were negative. These data allowed us to identify ∼40 novel and diverse virus genomes or genome fragments that contained an RdRp domain. We observed extensive sequence divergence of the novel RdRp domains, most sharing 25–40% amino acid identity with previously described RNA viruses (Table 2).

The most complete set of the RNA viruses was obtained from the springtails, due to the largest number of the transcriptomes. All novel RNA viruses were compared with the known viruses in the NCBI databases. In such a way, we obtained information about the RNA virus family/clade they belong to and their similarity to the already known viruses. To infer their phylogenetic position in the particular RNA viral clade, we used ML phylogenetic analysis. Phylogenies of basal hexapod RNA viruses (Figs. 1–3) demonstrated that they belong to numerous RNA viral clades. The RNA viral genomes of basal hexapods have genome organizations that are very similar or differ only slightly from the winged insect representatives (Figs. S11–S16). Since the sample processing for the preparation of transcriptomic and genomic libraries of basal hexapods involved entire individuals, a substantial proportion of the viruses discovered here might be associated with undigested food, gut microflora or parasites that exist within the organisms investigated. However, homology searching and phylogenies show that basal hexapod associated RNA viruses are most closely related to the insect viruses. It should be noted that the RNA virus sequences identified in the analysed transcriptomes of basal hexapods are their "putative" viruses and specific experiments should be carried out to prove that these viruses are indeed replicating in these arthropod species.

### Basal hexapods possess quite a diverse RNA virome

We found that basal hexapods possess representatives of 14 out of 24 RNA viral clades (Table 3). Such RNA virome diversity is higher than that of insect orders Blattodea (5/24), Dermaptera (6/24), Orthoptera (8/24), Lepidoptera (8/24) and Coleoptera (9/24). The only insect orders with similar or higher diversity of their RNA viromes are Odonata (12/24), Hemiptera (13/24) and Diptera (17/24) (Shi et al., 2016).

Basal hexapods possess three of the six known dsRNA viral clades: Reo, Partiti-Picobirna, and Toti-Chryso (Fig. 1). We found the first basal hexapod reovirus in *Anurida maritima*. Although this genome is partial, it is segmented. Six segments of reovirus were found and encode RdRp (VP1), VP2, VP3, VP4, VP5 and VP10 proteins (Data S1). This reovirus is quite divergent and shows less than 25% identity in the RdRp with the described reoviruses. It is most closely related to coltiviruses, significantly extending their host range (Fig. 1A). We found endogenized partitiviruses in the Machilis genome, they show 56% amino acid

**Table 1** **List of analysed basal hexapods in the NCBI transcriptome (TSA) and genome (WGS) databases.** Transcriptomes and genomes that were positive for RNA viruses are marked in blue.

| Basal hexapod lineage | Species | Transcriptome | Number of linear transcribed-RNAs | Genome |
|---|---|---|---|---|
| **Collembola** | *Anurida maritima* | GAUE02000000 | 22,076 | |
| | *Sminthurus viridis* | GATZ00000000 | 32,669 | |
| | *Tetrodontophora bielanensis* | GAXI00000000 | 46,137 | |
| | *Folsomia candida* | GAMN00000000 | 38,102 | LNIX00000000 |
| | *Orchesella cincta* | GAMM00000000 | 32,460 | LJIJ00000000 |
| | *Pogonognathellus sp.* | GATD00000000 | 37,079 | |
| | *Holacanthella duospinosa* | GFPE00000000 | 86,369 | NIPM00000000 |
| | *Sinella curviseta* | GGYG00000000 | 27,976 | RBVU00000000 |
| **Zygentoma** | *Atelura formicaria* | GAYJ00000000 | 51,705 | |
| | *Thermobia domestica* | GASN00000000 | 68,388 | |
| | *Tricholepidion gertschi* | GASO00000000 | 49,924 | |
| **Diplura** | *Catajapyx aquilonaris* | | | JYFJ02000000 |
| | *Occasjapyx japonicus* | GAXJ00000000 | 26,221 | |
| | *Campodea augens* | GAYN00000000 | 64,149 | |
| | *Megajapyx sp. UVienna-2012* | SRR400673 | 57,602 | |
| **Monocondylia** | *Machilis hrabei* | GAUM00000000 | 44,661 | QVQU01000000 |
| | *Meinertellus cundinamarcensis* | GAUG00000000 | 56,838 | |

identity with the Culex mosquito partitivirus. Their RdRp has a surprisingly well conserved coding capacity. A partial sequence of the partitivirus RdRp was found in the transcriptome of Thermobia (Zygentoma); it has 37% amino acid identity with the Hubei partiti-like virus 10 (Fig. 1B). Toti-Chryso clade has a single representative in the transcriptomes of basal hexapods, in the Tetrodontophora springtail. We obtained only an RdRp fragment, and a few coat proteins. In the Toti-Chryso tree, the springtail representative groups together with the "diatom colony-associated dsRNA virus 10" (Fig. 1C). We found endogenized totiviruses in the Machilis genome.

Basal hexapods possess representatives of four clades of the negative-stranded RNA viruses: Mono-Chu, Bunya-Arena, Orthomyxo and Qinvirus (Fig. 2). In the Mono-Chu clade, we found only endogenized mononegaviruses in genomes of Machilis and Catajapyx that belong to chuviruses (Fig. 2A), nyamiviruses and rhabdoviruses (Data S1). An endogenized Bunyavirus nucleoprotein was found in the genome of Machilis (Monocondylia) and shows 20–30% amino acid identity with phleboviruses (Data S1). Especially interesting was the discovery of highly divergent representatives of orthomyxoviruses in Atelura and Catajapyx (Fig. 2B). Endogenous Catajapyx orthomyxovirus is represented only by the PB1 protein. These two novel orthomyxovirus PB1 proteins show just 30% identity with the known orthologs. In Atelura, we found a nearly complete orthomyxoviral genome encoding five of the six segments (PB1, PB2, PA, envelope and nucleoprotein) (Fig. S11). We found a full-length representative of the Qinvirus clade in springtails, in the Anurida. This sequence is quite divergent from the others reported recently (Shi et al., 2016), showing just 25% identity in the RdRp region. Anurida qinvirus extends the host range of this rare viral clade from a few protostomes to

Ott Rutar and Kordis (2020), *PeerJ*, DOI 10.7717/peerj.8336

**Table 2  List of RNA viruses discovered in basal hexapods.**

| Putative host | NCBI accession number | Virus genome length (bp) | Length of RdRp (aa) | Virus classification (clade) | Closest relative | Amino acid identity (%) | E value | Query cover (%) |
|---|---|---|---|---|---|---|---|---|
| *Anurida maritima* | GAUE02021853 | 9,648 | 3,144 | Picorna-Calici | Thika virus | 22 | $2e^{-49}$ | 60 |
| *Anurida maritima* | GAUE02014165 | 8,391 | 2,106 | Picorna-Calici | Carfax virus | 32 | 0.0 | 83 |
| *Atelura formicaria* | GAYJ02042604 | 5,717 | 1,867 | Picorna-Calici | Hubei picorna-like virus 53 | 27 | $8e^{-53}$ | 35 |
| *Meinertellus cundinamarcensis* | GAUG02039188 | 5,753 | 1,885 | Picorna-Calici | Mayfield virus 1 | 38 | $6e^{-92}$ | 48 |
| *Occasjapyx japonicus* | GAXJ02019692 | 7,941 | 2,616 | Picorna-Calici | Hubei picorna-like virus 48 | 26 | $1e^{-69}$ | 30 |
| *Tetrodontophora bielanensis* | GAXI02037733 | 5,471 | 1,773 | Picorna-Calici | Kinkell virus | 35 | $8e^{-167}$ | 89 |
| *Campodea augens* | GAYN02051149 | 2,753 | 861 | Picorna-Calici | Baker virus | 32 | $1e^{-89}$ | 73 |
| *Sminthurus viridis* | GATZ02022882 | 10,627 | 2,546 | Hepe-Virga | Big Cypress virus | 29 | $5e^{-97}$ | 58 |
| *Anurida maritima* | GAUE02021637 | 9,420 | 2,837 | Hepe-Virga | Negev virus | 32 | $1e^{-137}$ | 52 |
| *Campodea augens* | GAYN02051120 | 2,383 | 216 | Hepe-Virga | Hubei virga-like virus 11 | 48 | $1e^{-54}$ | 95 |
| *Tetrodontophora bielanensis* | GAXI02034785 | 1,239 | 237 | Hepe-Virga | Hibiscus green spot virus 2 | 41 | $9e^{-33}$ | 71 |
| *Holacanthella duospinosa* | GFPE01052446 | 7,680 | 2,457 | Hepe-Virga | Hubei Beny-like virus 1 | 41 | 0.0 | 54 |
| *Atelura formicaria* | GAYJ02032904 | 2,190 | 529 | Tombus-Noda | Cushing virus | 43 | $1e^{-140}$ | 97 |
| *Tricholepidion gertschi* | GASO02037726 | 1,773 | 577 | Tombus-Noda | Hubei mosquito virus 4 | 28 | $6e^{-22}$ | 67 |
| *Tetrodontophora bielanensis* | GAXI02021960 | 1,133 | 377 | Narna-Levi | Wilkie narna-like virus 2 | 43 | $4e^{-82}$ | 99 |
| *Anurida maritima* | GAUE02014037 | 5,993 | 1,888 | Qinvirus | Hubei qinvirus-like virus 1 | 31 | 0.0 | 85 |
| *Atelura formicaria* | GAYJ02033071 | 2,485 | 803 | Orthomyxo | Sanxia Water Strider Virus 3 | 31 | $4e^{-107}$ | 97 |
| *Catajapyx aquilonaris* | JYFJ01081229 | – | 251 | Orthomyxo | Jingshan Fly Virus 1 | 46 | $9e^{-67}$ | 98 |
| *Machilis hrabei* | QVQU01083516 | 7,956 | 1,392 | Mono-Chu | Tacheng Tick Virus 6 | 28 | $2e^{-99}$ | 75 |
| *Machilis hrabei* | QVQU01249695 | 8,568 | 941 | Mono-Chu | Hubei chuvirus-like virus 4 | 40 | 0.0 | 96 |
| *Anurida maritima* | GAUE01055186 | 4,248 | 1,409 | Reo | Shelly headland virus | 35 | 0.0 | 98 |
| *Thermobia domestica* | GASN02036638 | 601 | 194 | Partiti-Picobirna | Hubei partiti-like virus 10 | 37 | $1e^{-28}$ | 100 |
| *Machilis hrabei* | QVQU01337473 | 3,397 | 478 | Partiti-Picobirna | Partitivirus-like Culex mosquito virus | 56 | $4e^{-177}$ | 90 |
| *Tetrodontophora bielanensis* | GAXI02022882 | 1,303 | 434 | Toti-Chryso | Diatom colony associated dsRNA virus 11 | 38 | $2e^{-84}$ | 99 |

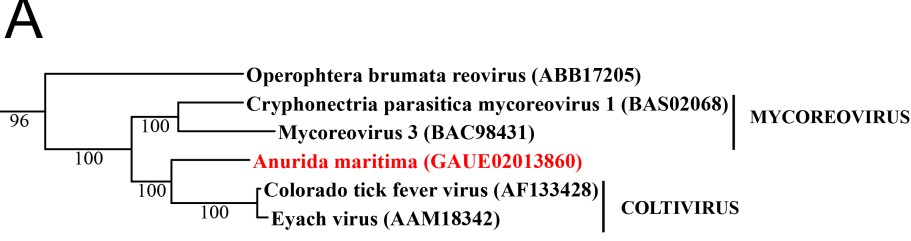

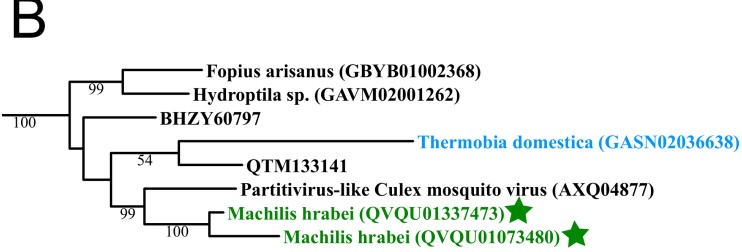

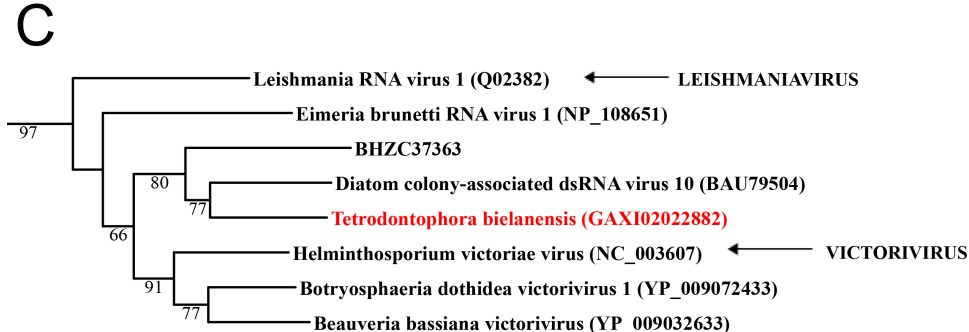

**Figure 1  Double stranded RNA viruses.** These midpoint-rooted, maximum-likelihood trees were inferred from viral RdRp protein sequences. The names of the viruses are marked with different colours based on their host taxonomy; springtails (Collembola) are red, Diplura are violet, Monocondylia are green, and Zygentoma are blue. The star symbol denotes host taxa that contain endogenous viral elements (EVEs). Sequences from the *Shi et al. (2016)* have the same unique accession numbers as in the original publication. Tree (A) reoviruses from the Reo clade; tree (B) partitiviruses from the Partiti-Picobirna clade; tree (C) totiviruses from the Toti-Chryso clade. Complete trees are provided in Figs. S1 to S3.

the basal hexapods (Fig. 2C). A complete genome of the Anurida qinvirus is 7,722 bp long, which represents a normal size for qinviruses. It is encoded in two segments, the larger one (5993 bp long) encodes RdRP, while the smaller one (1,729 bp long) encodes the putative structural protein that is homologous only to the Wuhan insect virus 15 (Fig. S12).

In basal hexapods, we found representatives of seven out of 12 positive-stranded RNA viral clades: Picorna-Calici, Hepe-Virga, Narna-Levi, Tombus-Noda, Flavi, Luteo-Sobemo and Permutotetra (Fig. 3). The largest diversity of the positive-stranded RNA viruses was found in the Picorna-Calici clade as the representatives of four picorna lineages were found –dicistrovirus (in Diplura only), iflavirus (in Diplura and Collembola), Kelp fly (in Zygentoma and Collembola) and Nora-like viruses (in Monocondylia only) (Fig. 3A). Few

**Table 3  RNA viromes in basal hexapod lineages.** The presence of RNA viral clade is marked with the black dot.

| RNA viral clade | Basal hexapods | Collembola | Diplura | Zygentoma | Monocondylia |
|---|:---:|:---:|:---:|:---:|:---:|
| Birna | | | | | |
| Partiti-Picobirna | ● | | | ● | ● |
| Reo | ● | ● | | | ● |
| Toti-Chryso | ● | ● | | | ● |
| Hypo | | | | | |
| Cystovir | | | | | |
| Bunya-Arena | ● | | | | ● |
| Mono-Chu | ● | | ● | | ● |
| Ophio | | | | | |
| Orthomyxo | ● | | ● | ● | |
| Qinvirus | ● | ● | | | ● |
| Yuevirus | | | | | |
| Hepe-Virga | ● | ● | ● | | |
| Luteo-Sobemo | ● | ● | | | |
| Narna-Levi | ● | ● | | | |
| Picorna-Calici | ● | ● | ● | ● | ● |
| Nido | | | | | |
| Tombus-Noda | ● | | | ● | ● |
| Weivirus | | | | | |
| Astro-Poty | | | | | |
| Flavi | ● | ● | | | |
| Permutotetra | ● | | ● | | |
| Yanvirus | | | | | |
| Zhaovirus | | | | | |
| | 14/24 | 8/24 | 5/24 | 4/24 | 8/24 |

selected picornaviral genomes are shown in the Fig. S13. We discovered Hepe-Virga clade in Collembola and Diplura, where Negev-like viruses were prevailing. In the phylogenetic analysis of the Negevirus group, we included diverse representatives in basal hexapods, extending the host range of this RNA viral clade (Fig. 3B). The genome of the Sminthurus negev-like virus is quite large (~10,6 kb). In addition to the RdRp-encoding ORF, it possesses additional ORFs with typical negevirus conserved protein domains (Fig. S14). We also found a complete Benji-like virus in Holacanthella (Collembola) transcriptome (Data S1). This virus is 45% identical in the core RdRp domain with the Hubei Beny-like virus 1, which was the first known metazoan benyi-like virus and was found only in Diptera (*Shi et al., 2016*). Its genome is 7,680 bp long and encodes a single ORF with 2,457 amino acids. The comparison of both metazoan benyi-like viruses showed that the springtail representative possesses a large region (between amino acids 850 and 1,969) that is absent in dipteran benyi-like virus. A single incomplete narnavirus genome (1,133 bp long) was found in springtails (Fig. 3C) and possesses a typical genome organization of narnaviruses (Fig. S14). Two representatives of the Tombus-Noda clade were found in Zygentoma, in

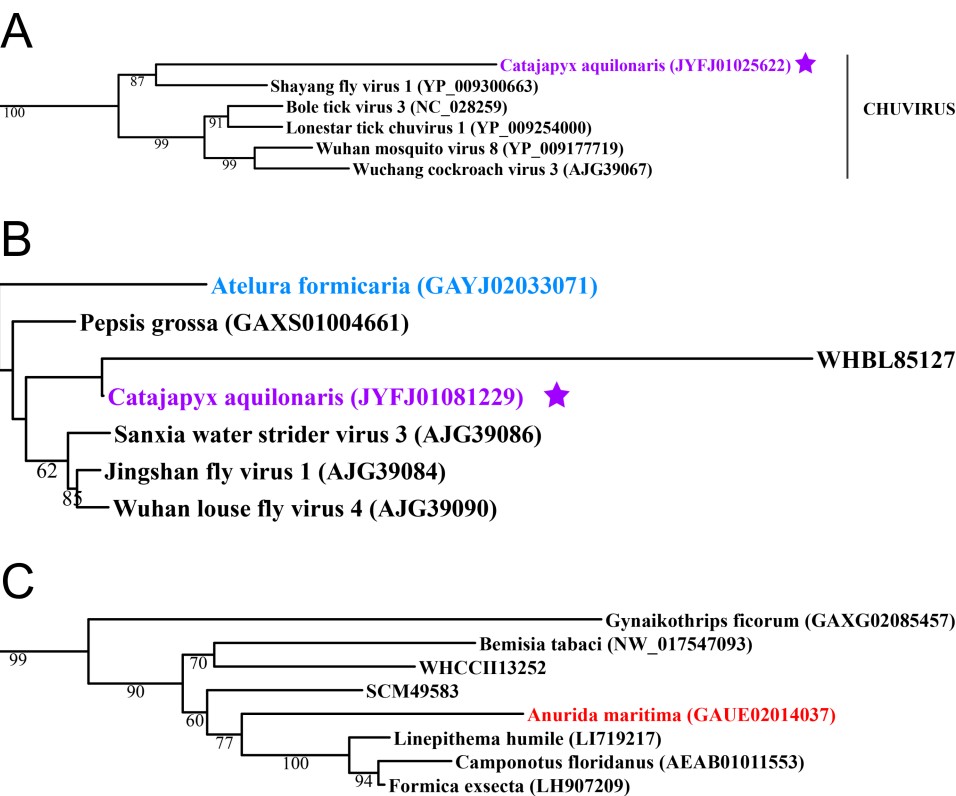

**Figure 2** **Negative-stranded RNA viruses.** Midpoint-rooted, maximum-likelihood trees were inferred from viral RdRp protein sequences. The names of the viruses are marked with different colours based on their host taxonomy; springtails (Collembola) are red, Diplura are violet, Monocondylia are green, and Zygentoma are blue. The star symbol denotes host taxa that contain endogenous viral elements (EVEs). Sequences from the *Shi et al. (2016)* have the same unique accession numbers as in the original publication. Tree (A) chuvirus from the Mono-Chu clade; tree (B) viruses belonging to Orthomyxo clade; tree (C) viruses belonging to Qinvirus clade. Complete trees are provided in Figs. S4 to S6.

Atelura and Tricholepidion transcriptomes (Fig. 3D). We found incomplete genomes of Tombus-like viruses in Zygentoma only (Fig. S16). In the Flavi clade, we found a number of fragments of jingmenvirus in a Sminthurus springtail, they encode both NS3 and NS5 proteins (Data S1). In a dipluran Megajapyx, we found few permutotetravirus fragments of the RdRp that show up to 59% amino acid identity with Hubei permutotetra-like virus 9 (Data S1). In the springtail Holacanthella, we found a fragment of sobemo-like virus RdRp that shows 47% amino acid identity with Hubei sobemo-like virus 17. In the transcriptome of the same species, we found sobemo-like capsid that shows 33% identity with Hubei sobemo-like virus 19. In the transcriptome of the Pogonognathellus springtail, we also found sobemo-like capsid (encodes viral-coat domain) that shows 34% identity with bat sobemovirus (Data S1).

## Endogenous viral elements are not rare in basal hexapods

A considerable number of endogenous viral elements (EVEs) was discovered in insect genomes (*Shi et al., 2016*), which is an indication of past infection events (*Holmes, 2011*;

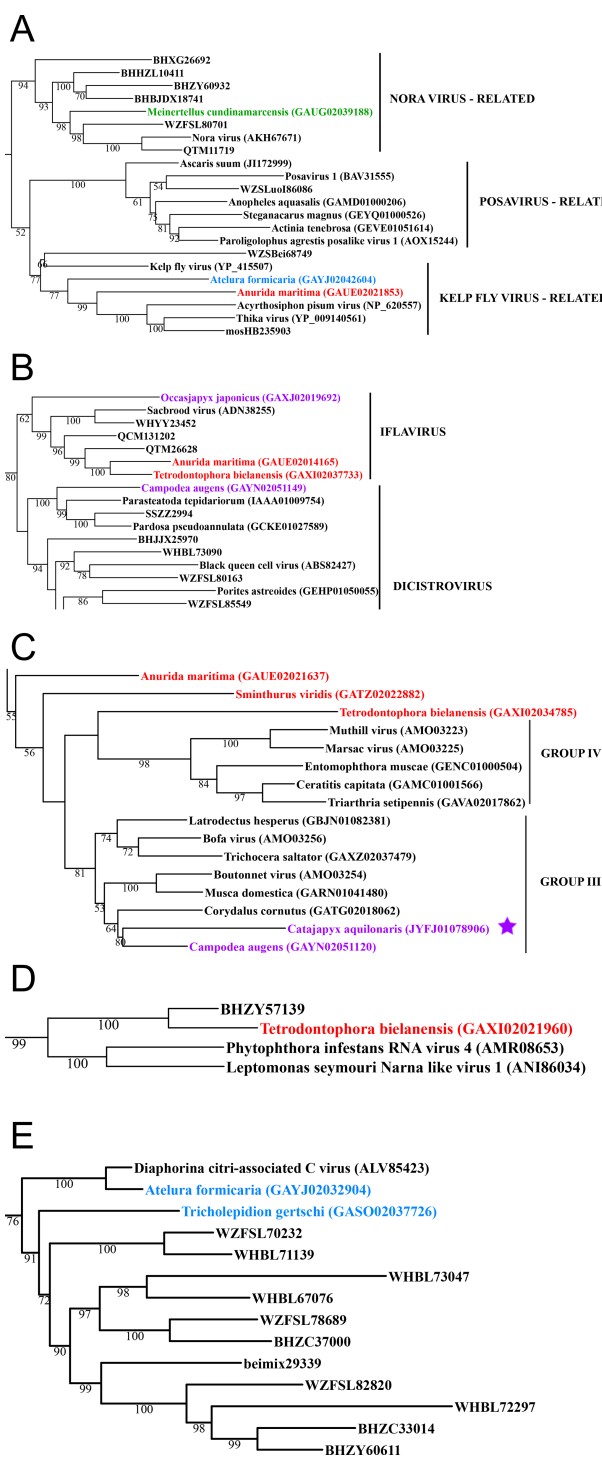

**Figure 3 Positive-stranded RNA viruses.** Midpoint-rooted, maximum-likelihood trees were inferred from viral RdRp protein sequences. The names of the viruses are marked with different colours based on their host taxonomy; springtails (Collembola) are red, Diplura are violet, Monocondylia are green, and Zygentoma are blue. The star symbol denotes host taxa that contain (continued on next page...)

**Figure 3 (...continued)**
endogenous viral elements (EVEs). Viruses previously described as EVEs are also marked. Sequences from the *Shi et al. (2016)* have the same unique accession numbers as in the original publication. Tree (A) viruses belonging to Nora virus, Posavirus and Kelp fly virus lineages from Picorna-Calici clade; tree (B) viruses belonging to iflavirus and dicistrovirus lineages from Picorna-Calici clade; tree (C) viruses belonging to the Negeviruses from the Hepe-Virga clade; tree (D) viruses belonging to the Narnaviruses from the Narna-Levi clade; tree (E) viruses belonging to the tombus-like viruses from the Tombus-Noda clade. Complete trees are provided in Figs. S7 to S10.

**Table 4  Endogenous viral elements in basal hexapod genomes.** The presence of RNA viral clade is marked with the black dot.

| RNA viral clade | Monocondylia | Diplura | basal hexapods | Arthropoda |
|---|---|---|---|---|
| Hepe-Virga | | ● | ● | ● |
| Luteo-Sobemo | | | | ● |
| Narna-Levi | | | | ● |
| Bunya-Arena | ● | | ● | ● |
| Mono-Chu | ● | ● | ● | ● |
| Orthomyxo | | ● | ● | ● |
| Nido | | | | ● |
| Partiti-Picobirna | ● | | ● | ● |
| Picorna-Calici | | | | ● |
| Reo | ● | | ● | ● |
| Tombus-Noda | ● | | ● | ● |
| Toti-Chryso | ● | | ● | ● |
| Qinvirus | ● | | ● | ● |
| | 7/24 | 3/24 | 9/24 | 13/24 |

*Feschotte & Gilbert, 2012*; *Aiewsakun & Katzourakis, 2015*). We searched for potential EVEs in all available basal hexapod genomes. We analysed RdRp proteins as well as numerous additional structural proteins, such as nucleo- and glyco-proteins, for representatives of all metazoan RNA viral families. EVEs in basal hexapods came from nine RNA viral clades: from Mono-Chu, Orthomyxo, Qin, Partiti-Picobirna, Reo, Tombus-Noda, Hepe-Virga, Bunya-Arena and Toti-Chryso (Figs. 1–3; Data S1). Since the RNA viromes of basal hexapods are diverse, it is interesting that they possess EVEs only in Diplura and Monocondylia genomes. Although arthropods possess EVEs for 13 RNA viral clades (Hepe-Virga, Luteo-Sobemo, Narna-Levi, Bunya-Arena, Mono-Chu, Orthomyxo, Nido, Partiti-Picobirna, Picorna-Calici, Reo, Tombus-Noda, Toti-Chryso and Qinvirus) (*Shi et al., 2016*), we found that the amount and the diversity of EVEs in basal hexapods is similar to them (Table 4). As expected, given their endogenous status, most of these sequences are only fragments of the parent virus genome (Figs. 1–3; Data S1). All EVEs in basal hexapods are integrated in random genomic loci in different species. In the vicinity of EVEs in basal hexapods no retroposon elements can be found. Our analysis demonstrated that EVEs are not rare in the genomes of some basal hexapods and have been generated by multiple independent integration events.

## Composition and abundance of RNA viruses in viromes of basal hexapods

The comparison of basal hexapod RNA virome composition with that of insects (*Shi et al., 2016*) demonstrated some similarities and many differences. In both cases, Picorna-Calici clade is the largest. Hepe-Virga is the major additional clade in basal hexapods. The other RNA viral clades are much less abundant in the total RNA virome of basal hexapods, such as the Mono-Chu, Tombus-Noda, Narna-Levi, Partiti-Picobirna, Luteo-Sobemo, Orthomyxo, Reo, Toti-Chryso, Flavi, Permutotetra, Bunya-Arena and Qinvirus. Positive-stranded RNA viruses are prevailing in their RNA virome, while negative-stranded and double-stranded RNA viruses are less abundant in basal hexapods (Figs. 1–3). It was demonstrated that the abundance and composition of RNA viromes are obviously phylum-specific (*Shi et al., 2016*). While some RNA viral clades are much more abundant in winged insects, they are quite rare in basal hexapods. The reasons for such differences could be effects of biased sampling, depth and size of the RNASeq libraries, or real differences in the amount of some RNA viral clades. The majority of hexapods possess very similar patterns of RNA virome composition –few major RNA viral clades and numerous minor clades with limited distribution (*Shi et al., 2016*).

## Comparison of winged insect (Pterygota) and basal hexapod RNA viromes

The basal position of apterygote hexapods in the hexapod tree (*Misof et al., 2014*) is important for understanding the origin and evolution of the insect-specific RNA viruses. We can compare novel basal hexapod RNA viruses with diverse relatives from winged insects. In such a way, we can trace the changes in the RNA viromes, originations of particular RNA viral families etc. (Table 5). It is obvious that basal hexapod and winged insect RNA viromes are similar, where insects collectively possess four viral clades more (18 of the 24 RNA viral clades in total) (*Shi et al., 2016*). It should be noted that all previously discovered insect RNA viruses were involved in the phylogenetic analysis of the invertebrate RNA virosphere (*Shi et al., 2016*).

dsRNA virome of the basal hexapods is represented by three RNA viral clades, while winged insects possess representatives of five viral clades. Some insect orders are without (Orthoptera), with a single (Lepidoptera, Dermaptera and Blattodea) or with just two dsRNA viral clades (Coleoptera). Insect orders with three or four RNA viral clades are Odonata, Hemiptera and Diptera. As evident from the Table 5, there are differences between the insect orders in the presence/absence of the particular dsRNA viral clade. A similar situation was observed in negative-stranded RNA viromes where basal hexapods possess representatives of four RNA viral clades. In insects, five negative-stranded RNA viral clades are present, but with unequal distribution patterns in diverse insect orders. Some of these RNA viral clades are diverse and rich (Mono-Chu and Bunya-Arena clades), while others are moderate (Orthomyxo) or very small (Ophio and Qinvirus). Some insect orders possess a single (Coleoptera), two (Lepidoptera and Dermaptera) or three RNA viral clades (Orthoptera, Blattodea, Odonata and Hemiptera). Diptera is the only insect order that possesses five out of six negative-stranded RNA viral clades. Until now, dipterans

Ott Rutar and Kordis (2020), *PeerJ*, DOI 10.7717/peerj.8336

**Table 5  Comparison of insect and basal hexapod RNA viromes.** The presence of RNA viral clade is marked with the black dot. Insect and crustacean RNA viral data are from *Shi et al. (2016)*. The data for specific insect orders (Coleoptera to Diptera) are included.

| RNA viral clade | Basal hexapods | insects | crustaceans | Coleoptera | Lepidoptera | Orthoptera | Dermaptera | Blattodea | Odonata | Hemiptera | Diptera |
|---|---|---|---|---|---|---|---|---|---|---|---|
| Birna | | • | | | | | | | | | • |
| Partiti-Picobirna | • | • | • | • | | | • | | • | • | • |
| Reo | • | • | • | • | • | | | | • | • | • |
| Toti-Chryso | • | • | • | | | | | • | • | • | • |
| Hypo | | • | | | | | | | | | |
| Cystovir | | | | | | | | | | | |
| Bunya-Arena | • | • | • | | • | • | • | | • | • | • |
| Mono-Chu | • | • | • | • | • | • | | • | • | • | • |
| Ophio | | • | | | | | | | | | • |
| Orthomyxo | • | • | • | | | • | • | • | • | • | • |
| Qinvirus | • | • | • | | | | | | | | • |
| Yuevirus | | | • | | | | | | | | |
| Hepe-Virga | • | • | • | • | • | | | | • | • | • |
| Luteo-Sobemo | • | • | • | • | • | • | | | • | • | • |
| Narna-Levi | • | • | • | • | | | • | • | • | • | • |
| Picorna-Calici | • | • | • | • | • | • | • | | • | • | • |
| Nido | | • | • | | | | | | | | • |
| Tombus-Noda | • | • | • | • | • | • | • | | • | • | • |
| Weivirus | | | | | | | | | | | |
| Astro-Poty | | | • | | | | | | | | |
| Flavi | • | • | • | | | • | | | | • | • |
| Permutotetra | • | • | • | • | • | • | | | • | • | • |
| Yanvirus | | | • | | | | | | | | |
| Zhaovirus | | | • | | | | | | | | |
| | 14/24 | 18/24 | 19/24 | 9/24 | 8/24 | 8/24 | 6/24 | 5/24 | 12/24 | 13/24 | 17/24 |

were the only insect order that possesses Quinvirus (*Shi et al., 2016*). Here, we show that qinviruses are indeed more widespread among insect orders (Fig. 2). The positive-stranded RNA virome in basal hexapods is represented by seven viral clades while winged insects possess eight clades out of twelve. In contrast to the basal hexapods, winged insects possess more diversity inside the RNA viral clades of the positive-stranded RNA virome. A number of insect orders possess a smaller number of positive-stranded RNA viral clades, such as Blattodea (1), Dermaptera (3), Lepidoptera (5), Orthoptera (5), Coleoptera (6) and Odonata (6). Hemiptera possess seven clades, while Diptera possess eight positive-stranded RNA viral clades. As in the double-stranded and negative-stranded RNA viromes, the distribution of positive-stranded RNA viral clades differs among insect orders. Only some of the RNA viral clades are widespread in hexapods, such as Picorna-Calici, Tombus-Noda, Mono-Chu, Bunya-Arena, Orthomyxo and Permutotetra. All other RNA viral clades have a much more limited distribution.

## Horizontal virus transfer is very rare in basal hexapods

Many insects are known vectors for the dissemination of RNA viruses, such as mosquitoes and many plant pests (e.g., thrips, whiteflies, lepidopterans, coleopterans and scale insects) (*Whitfield, Falk & Rotenberg, 2015*; *Rückert & Ebel, 2018*). Since springtails (Collembola) are very abundant physical decomposers of plant and fungal material, there is a possibility of transfer of plant or fungal viruses into them. Springtails could potentially act as vectors of plant or fungal RNA viruses. However, the analysis of springtail transcriptomes and genomes showed that horizontal virus transfer (HVT) is extremely rare among them. We found only a single short fragment of a plant RNA virus in the springtail transcriptome, which could be present in ingested plant material infected with this virus. This was the alfalfa mosaic virus (AMV, Bromoviridae), found in the Holacanthella transcriptome (GFPE01073448, 340 bp long fragment, 99% amino acid identity to AMV). We were unable to find any sign of HVT in any other basal hexapod lineage.

## DISCUSSION

The research on insect viruses has been very intensive in the last decade (*Bonning, 2019*). This has been mostly due to the application of metagenomic and metatranscriptomic approaches (*Junglen & Drosten, 2013*; *Liu, Chen & Bonning, 2015*; *Obbard, 2018*). A major breakthrough has been achieved recently when a large-scale analysis of invertebrate RNA virosphere has been published (*Shi et al., 2016*). This study extends their previous studies on negative-stranded RNA viruses (*Li et al., 2015*) and flavivirus-like proteins (*Shi et al., 2015*). A large proportion of novel data in these three studies was obtained from diverse insect orders. The novel picture has revealed quite large differences in RNA virus diversity and their distribution patterns among diverse insect orders. However, there are still numerous arthropod groups that were not included in the extensive analyses of RNA viral diversity. One of these groups are basal hexapods. Until now, the only reported basal hexapod RNA virus was an amalgavirus, which was found in Tetrodontophora springtail but very likely originated from the microsporidian pathogen (*Pyle, Keeling & Nibert, 2017*).

For that reason, we analysed RNA viruses in publicly available transcriptomes and genomes for basal hexapods.

Here, we demonstrated that basal hexapods possess 14 out of 24 RNA viral clades, which are the following: Reo, Partiti-Picobirna, Toti-Chryso, Mono-Chu, Bunya-Arena, Orthomyxo, Qinvirus, Picorna-Calici, Hepe-Virga, Narna-Levi, Tombus-Noda, Luteo-Sobemo, Permutotetra and Flavi. Such RNA virome diversity is similar to that of insects and is even higher than in some large insect orders (Table 5). In this study, we uncovered some highly divergent viruses that have only 25–30% amino acid identity in their RdRps with the known RNA viruses. These highly divergent basal hexapod RNA viruses are qinvirus, reovirus, orthomyxovirus and negev-like viruses. Genome organizations of the basal hexapod RNA viruses are very similar to the winged insect representatives (Figs. S11–S16) (*Shi et al., 2016*). In this study, we extended the host range for some rare RNA viruses, such as qinviruses and coltivirus. In Picornavirales, we found representatives of dicistroviruses, iflaviruses, Nora-like and Kelp-fly viruses. In this way, we obtained novel representatives of mostly insect-specific picornaviruses. Iflaviruses seem to be prevalent among basal hexapod picornaviruses. Transcriptome libraries made by selecting polyadenylated RNAs might substantially bias against certain types of RNA viruses without poly-A genomes. However, this was definitely not the case in basal hexapod transcriptomes, since we observed besides the three polyadenylated RNA viral clades (Orthomyxo, Hepe-Virga, Picorna-Calici) also nine nonpolyadenylated RNA viral clades (Partiti-Picobirna, Reo, Toti-Chryso, Qinvirus, Luteo-Sobemo, Narna-Levi, Tombus-Noda, Flavi and Permutotetra).

We believe that our approach was sensitive enough to find some of the most divergent arthropod RNA viruses. Due to the high divergence of basal hexapod RNA viruses, we used several representatives of the particular RNA viral family or clade as queries. As a rule, we used three representatives of the RNA viral clade, as defined by *Shi et al. (2016)*, on both extremes and in the middle of the tree. Instead of the default parameters in homology searching with TBlastN, we also used some modified parameters to find remote homologs or very divergent RNA viruses. However, in both cases we obtained the same set of RNA viruses and no extremely divergent viruses. Despite this, the novel basal hexapod RNA viruses are among the most divergent arthropod RNA viruses; very often they share just 22–40% identity with the already described RNA viruses. It should be noted that the closest relatives of the novel basal hexapod RNA viruses are always from the arthropod hosts.

EVEs in basal hexapods came from nine RNA viral clades: Mono-Chu, Orthomyxo, Qin, Partiti-Picobirna, Reo, Tombus-Noda, Hepe-Virga, Bunya-Arena and Toti-Chryso (Figs. 1–3; Data S1). Since the RNA viromes of basal hexapods are diverse, it is interesting that they possess EVEs only in Diplura and Monocondylia genomes. Since whole organisms were used for the preparation and sequencing of genomic DNA, there is a big chance that a number of the putative EVEs are indeed RNA viruses associated with the basal hexapod hosts. However, most of the EVE sequences in basal hexapod genomes are highly fragmented (Data S1), as expected for their endogenous status. No retrotransposon elements can be found in the vicinity of EVEs in basal hexapods. Little is known about the underlying molecular mechanisms, but sequence signatures at the EVE–host genome

junction point to retroposition events, suggesting involvement of the enzymatic machinery encoded by retrotransposons residing in the host genome (*Feschotte & Gilbert, 2012*).

Basal hexapods were the earliest splits of hexapod lineages (*Misof et al., 2014*). The sister group of hexapods are crustaceans with incredibly diverse RNA viromes, especially in marine crustaceans. Land crustaceans (e.g., isopods) have a much lower abundance and diversity of their RNA viromes (*Shi et al., 2016*). The situation seems to be similar in basal hexapods, where the diversity of the RNA virome is quite high, but the abundance of the RNA viruses is lower than in some large insect orders (e.g., in dipterans). Ecology (soil and plant material decomposers) and the extremely high abundance of springtails (Collembola) (*Rusek, 1998*) offer the possibility to act as vectors in HVT (*Shi et al., 2016*; *Li et al., 2015*; *Dolja & Koonin, 2018*; *Blanc & Gutierrez, 2015*). However, their RNA virome and RNA viruses do not show any significant amount of HVT.

Many of the RNA viruses might not infect hexapods, but their parasites (*Grybchuk et al., 2018*). The problem of finding the true host in holobiont virome analysis has already been explained before (*Shi et al., 2016*; *Dolja & Koonin, 2018*). The source of the RNA viruses in the holobiont sequences might be undigested food, gut microflora or parasites that exist within the organisms investigated (*Shi et al., 2016*). Although some insect orders (mainly Diptera, Heteroptera and fleas) are infected with kinetoplastid parasites, it seems that basal hexapods are not their hosts. However, diverse gregarines (Apicomplexa, Alveolata) are known to be parasites of basal hexapods. At least six genera of gregarines parasitize diverse lineages of basal hexapods. Currently, no sequence data are available for basal hexapod-associated gregarines. Transcriptome data for gregarines are mostly from annelid or mollusk hosts. We checked the diversity of the RNA viruses in gregarine transcriptomes at the NCBI TSA Db and found that at least 13 RNA viral clades are associated with gregarines. These are Picorna-Calici, Hepe-Virga, Tombus-Noda, Flavi, Narna-Levi, Yanvirus, Astro-Poty, Mono-Chu, Bunya-Arena, Ophio, Qinvirus, Partiti-Picobirna and Toti-Chryso. A caveat should be taken into account since the contaminant contigs derived from gut cells of the animal host or other organisms in the gut may be present in these transcriptomes. Despite this, none of their RNA viruses is highly similar to any invertebrate RNA virus. All those RNA viruses are highly divergent; some are very likely novel metazoan representatives, while others may be genuine gregarine RNA viruses. Moreover, none of the gregarine RNA viruses is very similar to the basal hexapod RNA viruses. Homology searching and phylogenies have shown that the basal hexapod-associated RNA viruses are most closely related to the insect viruses. RNA viruses of the gregarine parasites also significantly differ from arthropod sequences. We think that shared parasites can assist in the HVT of RNA viruses between unrelated hosts. However, current data indicate that the HVT of RNA viruses in the basal hexapods is negligible. A much larger population sampling of springtails in nature could provide evidence about their role as potential vectors for the dissemination of viruses.

What are these viruses doing to their hosts? Although viruses are parasites, some of them might be mutualists or commensals and their impact on host fitness may be negligible (*Obbard, 2018*; *Cadwell, 2015*; *Virgin, 2014*; *Roossinck, 2011*). We were unable to determine whether the viruses identified here have any impact on host biology, including as agents of

disease. Despite this, it is clear that for many metazoans infection by multiple RNA viruses is likely to be the norm rather than the exception (*Shi et al., 2016*; *Shi et al., 2018*).

## CONCLUSIONS

Our study demonstrated that basal hexapods possess quite a diverse RNA virome and some highly divergent RNA viruses. Going forward, the 1KITE (http://www.1kite.org) and i5K (arthropodgenomes.org/wiki/i5K/) projects will generate numerous additional genomes and transcriptomes for understudied basal hexapods. These new data may provide additional insights into the RNA virome of the basal hexapod lineages.

## ACKNOWLEDGEMENTS

The authors thank Prof. Roger H. Pain for his critical reading of the manuscript. Our sincere thanks go to Dr. Jernej Šribar for his technical assistance.

### Funding

This work was supported by the Slovenian Research Agency grant P1-0207. The funders had no role in study design, data collection and analysis, decision to publish, or preparation of the manuscript.

### Grant Disclosures

The following grant information was disclosed by the authors:
Slovenian Research Agency: P1-0207.

### Competing Interests

The authors declare there are no competing interests.

### Author Contributions

- Sabina Ott Rutar and Dusan Kordis analyzed the data, conceived and designed the experiments, performed the experiments, prepared figures and/or tables, authored or reviewed drafts of the paper, and approved the final draft.

### Data Availability

Raw data is available in the Data S1.

The novel RNA viruses of basal hexapods are available at NCBI Transcriptome Shotgun Assembly (TSA): GAUE02014037.1, GAUE02011884.1, GAXI02022882.1, GAMM01008132.1, GAYJ02033071.1, GAYJ02032054.1, GAYJ02040263.1, GAYJ02033043.1, GAYJ02033073.1, GAUE02021853.1, GAYJ02042604.1, GAXI02021960.1, GAYN02051120.1, GAUE02021637.1, GATZ02022882.1, GAXI02034785.1, GAUG02039188.1, GAXJ02019692.1, GAUE02014165.1, GAXI02037733.1, GAYN02051149.1, GAYJ02032904.1, GASO02037726.1,

GFPE01052446.1, GAUE02013860.1, GAUE02012122.1, GAUE02012094.1, GAUE02011164.1, GAUE02012784.1, GAUE02013011.1, GAUE02012054.1, GASN02036638.1.

The EVEs of RNA viruses of basal hexapods are available at NCBI Whole Genome Shotgun: JYFJ01025622.1, JYFJ02009787.1, JYFJ00000000.2, QVQU01083516.1, QVQU01249695.1, QVQU01337473.1.

## Supplemental Information

Supplemental information for this article can be found online at http://dx.doi.org/10.7717/peerj.8336#supplemental-information.

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
