# Peer review of "Analysis of the RNA virome of basal hexapods"

_PeerJ, doi:10.7717/peerj.8336_

## Round 0.1 · original submission · Minor Revisions

Both reviewers sound positive; please address their points.

Reviewer 1 ·

Basic reporting

The work of Rutar & Kordis is interesting and timely. Analyzing viral diversity in different hosts is a very hot topic these days. I liked the manuscript and, in my opinion, it should be published after some modifications. I believe 3 ideas deserve to be discussed thoroughly:

1) Many of these viruses might infect not hexapods, but their parasites. Such examples have recently been reported by Fasel, Yurchenko & Beverley laboratories. In this sense, it may be necessary to cite some of their papers, e.g. Grybchuk et al PNAS USA and mBio, Akopyants et al, Genome Announc, etc This topic can also be discussed from the angle of HVT: it is easier to imagine this process via shared parasites.

2) Is author's approach sensitive enough? It may easily miss some divergent viruses - see the case of Ostravirus in one of the above mentioned papers.

3) Can retroposon elements be identified in the vicinity of EVEs?

Minot comment:
Sentences on lines 90-94 appear redundant.

Experimental design

The experimental design is adequate. Methods are described with sufficient detail & information to allow replication.

Validity of the findings

The findings are scientifically sound.

Additional comments

I am also curios whether EVEs are integrated in some random genomic loci in different species or not. This, probably, goes beyond the scope of this manuscript, but authors might include this info, if it is available.

·

Basic reporting

This study identified a number of sequences of novel RNA viruses in the transcriptomes of 16 species of basal hexapods which belonged to 4 evolutionary divergent phyla, which were available Transcriptome Shotgun Assembly at NCBI. Also, a number of endogenous viral elements derived from RNA viruses of different clades were identified in the genomes of 6 sequenced basal hexapod species.

Basal hexapods are poorly characterised arthropods despite their ecological importance due to their role in decomposition of plant and animal remains, therefore identification of putative hexapod viruses could be basis of future work on basal hexapod biology and pathology.

Experimental design

The RNA virus sequences were identified based on the amino acid homology with previously published RNA virus sequences (in particular RdRp) using TBLASTN.

Validity of the findings

To my knowledge, this is the first systematic screening of the published basic hexapod transcriptomes (and genomes) for the presence of RNA virus sequences. Identification of RNA-virus sequences was carried out using the appropriate approach.

Additional comments

The most important outcome of the analysis is the list of the newly identified virus sequences. Therefore the main text should include a table listing GenBank accession numbers for all 27 RNA virus sequences shown in “Supplementary Data: Nucleotide sequences of basal hexapod RNA viruses.” This summary table should also provide additional data (e.g. source basal hexapod transcriptome, viral sequence length, GenBank accession of the previously published virus sequence showing highest homology with the new virus sequence with BLASTP E-value, and possibly %% of coverage in viral genome (sections)).

Throughout the text: “RdRP” -> “RdRp”

Line 45: ”…viruses, some of which are present as endogenous genomic copies” - these are not viruses but virus-derived sequences in genome - change to “endogenous viral elements derived from RNA viruses ”

In all figure legends showing phylogenetic trees specify that these are amino acid alignments.

Line 147: “…, we collected all RdRP proteins translated from the virus sequence collections described above.” -> “…, we used RdRp protein sequences”.

Lines 178-182. It should be noted that the RNA virus sequences identified in the analysed transcriptomes of basal hexapods are their "putative" viruses and specific experiments should be carried out to prove that these viruses are indeed replicating in these arthropod species.

Lines 260-261 “Our analysis thus demonstrated that basal hexapods possess a diverse RNA virome that is richer than in numerous insect orders.”
Remove this claim, the number of RNA virus clades found in "Basal hexapods" is very similar to that of "Crustacea" and "Insects" (with all insect orders combined) - Fig. 4. Note that different clades of basal hexapods are more diverse than orders within Insecta (Gao et al (2008) Zoological Science 25: 1139–1145 doi:10.2108/zsj.25.1139 ), it is more correct to compare diversity of viruses infecting all Insecta orders combined with that of basal hexapods.

Table 4. Specify that "Coleoptera-Lepidoptera...- to - Diptera" are "Insects"

Line 303-304. "...basal hexapod RNA virome differ from the rich arthropod RNA virome" -
Analysis of only 16 basal hexapod transcriptomes already resulted in identification of viruses which belong to 14 clades of 24 found in other arthropods. It could be a case that viruses belonging to oath clades could be found with wider screening. this is not excluding possibility of "basal hexapod-specific" virus clades.

Line 342. alfalfa mosaic virus ( plant virus) RNA could be a present in ingested plant material infected with this virus. thesis definitely not a case of "horizontal virus transfer"

---

## Round 0.2 · accepted · Accept

All reviewers' concerns have been addressed.

Reviewer 1 ·

Basic reporting

no comment

Experimental design

no comment

Validity of the findings

no comment